# An Extra Set of Intelligent Eyes: Application of Artificial Intelligence in Imaging of Abdominopelvic Pathologies in Emergency Radiology

**DOI:** 10.3390/diagnostics12061351

**Published:** 2022-05-30

**Authors:** Jeffrey Liu, Bino Varghese, Farzaneh Taravat, Liesl S. Eibschutz, Ali Gholamrezanezhad

**Affiliations:** Keck School of Medicine, University of Southern California, Los Angeles, CA 90033, USA; liu501@usc.edu (J.L.); bvarghes@usc.edu (B.V.); farzanehtaravat6485@gmail.com (F.T.); liesl.eibschutz@med.usc.edu (L.S.E.)

**Keywords:** artificial intelligence, radiology, imaging, computed tomography, abdominal pain, GI trauma

## Abstract

Imaging in the emergent setting carries high stakes. With increased demand for dedicated on-site service, emergency radiologists face increasingly large image volumes that require rapid turnaround times. However, novel artificial intelligence (AI) algorithms may assist trauma and emergency radiologists with efficient and accurate medical image analysis, providing an opportunity to augment human decision making, including outcome prediction and treatment planning. While traditional radiology practice involves visual assessment of medical images for detection and characterization of pathologies, AI algorithms can automatically identify subtle disease states and provide quantitative characterization of disease severity based on morphologic image details, such as geometry and fluid flow. Taken together, the benefits provided by implementing AI in radiology have the potential to improve workflow efficiency, engender faster turnaround results for complex cases, and reduce heavy workloads. Although analysis of AI applications within abdominopelvic imaging has primarily focused on oncologic detection, localization, and treatment response, several promising algorithms have been developed for use in the emergency setting. This article aims to establish a general understanding of the AI algorithms used in emergent image-based tasks and to discuss the challenges associated with the implementation of AI into the clinical workflow.

## 1. Introduction

Spawned by the age of digital medical imaging and high throughput computing, Artificial intelligence (AI) is one of the major innovations of the healthcare sector, particularly in radiology [1,2,3]. On a simplistic level, AI can be defined as a field of science involved in the creation of systems that perform problem-solving tasks that require human intelligence [4]. Machine learning is a specific subset of AI that creates algorithms and statistical models that dictate the performance of AI systems in performing specific user-defined tasks without using explicit instructions. These methods rely on predefined engineered features derived from expert knowledge that quantify radiographic characteristics, including volume, shape, size, texture, intensity, and location. The most robust features are then selected and fed into statistical machine learning models to identify imaging biomarkers. Machine learning methods rely on patterns and inferences derived from the human-engineered data extracted from images to perform the task.

In recent years, the ongoing improvements in AI applications have focused on deep learning systems that scale with data, rather than the traditional machine learning methods based on predefined radiologic features. These deep learning systems encompass a specific machine learning approach that relies on artificial neural network (ANN) algorithms and contains hidden layers, such as convolutional neural networks (CNN), to process raw data and perform classification or detection tasks. By design, deep learning is a representation-learning method with multiple levels of representation. Thus, deep learning methods do not require human-engineered features such as inputs as they automatically extract features from images. This in turn reduces the need for the manual processing of images, a time-consuming endeavor that is oftentimes subject to operator bias [5]. Yet, many clinicians are unaware of the complex relationships inherent in the deep learning algorithms, rendering this approach difficult to accept within a clinical setting. While machine learning models may be more efficient in some cases (e.g., when the input data involve single-number metrics), deep learning models, although requiring substantially more input information, may outperform machine learning approaches with more complex data.

As emergency radiology confronts mounting expectations to deliver dedicated on-site service and the demand of increased imaging load and rapid report turnaround time, this field represents a promising avenue of entry for AI into radiology departments globally. Yet, there is a paucity of literature describing AI applications in this field, particularly with regard to the imaging of emergent abdominopelvic pathologies and the practical implications. Deep learning systems provide an opportunity to augment human decision making and improve efficiency and effectiveness, thus achieving a true enhancement in the quality of care [6,7]. Ultimately, this review will provide an overview of the AI applications for abdominopelvic imaging in emergency radiology according to pathologic classification: (1) diseases of the digestive tract, (2) trauma, and (3) abdominal aortic aneurysms.

## 2. Overview

In recent years, several promising algorithms in a variety of fields have been developed for use in the emergency setting (Table 1). For instance, AI algorithms demonstrated clinical utility in non-contrast, head computed tomography (CT) scans to detect hemorrhage, mass effect, hydrocephalus, and suspected acute infarct. Additional promising AI applications include the detection and classification of chest abnormalities in chest radiographs and CT scans; the identification and quantification of coronary artery calcification in CT scans; and fracture detection in orthopedic trauma. Several studies have also described the utility of AI in oncologic detection, localization, and treatment response. Yet, irrespective of the application, the primary advantage of AI implementation in radiology is the ability to act as a second opinion [8]. This has the potential to improve diagnostic accuracy, particularly in resource-limited settings. AI also has great utility in triaging patients as the algorithm can divide abnormal cases based on predetermined criteria that we can define, and the radiologist can then work off the sorted priority list. This has the potential to improve workflow efficiency, with faster turnaround results for complex cases and reduced overall workloads.

## 3. Diseases of the Digestive Tract

### 3.1. Small Bowel Obstruction

As small bowel obstruction is a common diagnostic cause of acute abdominal pain in the emergency setting, patients with high clinical suspicion oftentimes undergo abdominal radiography as one of the first-line screening tests. Three radiologic signs are highly pathognomonic for the detection of SBO: two or more air-fluid levels, air-fluid levels wider than 2.5 cm, and air-fluid levels differing more than 5 mm from one another in the same loop of bowel [12]. Yet, recent studies indicate that a direct correlation exists between SBO detection and radiologist experience [11]. Thus, AI-assisted detection may aid non-radiologists or junior radiology staff members in diagnosing this pathology [11].

Although difficult to comprehend, the mechanisms behind deep learning architecture can be investigated with the use of occlusion maps. Occlusion maps are developed by occluding areas in the original image with a low probability of detecting pathology. These maps can change the output probability of neural network models, making it easier to parse out relevant image details for classification when overlayed on the original image. The application of occlusion maps in neural network models of SBO detection appears to demonstrate the significance of dilated small bowel segments, but other specific SBO features may be universally distributed throughout the image [13]. In a single-institution pilot study of AI technology, transfer learning from a pre-trained neural network was conducted on a set of 3663 clinical supine abdominal radiographs. Four hundred and fifty-two images were classified by the transferred neural network as false positives, of which 94 images (21%) were considered as ileus and 50 images (11%) were considered low-grade bowel obstruction [13]. Following training, the neural network achieved an AUC of 0.849, and an observed sensitivity and specificity of 84% and 68%, respectively [13]. Follow-up studies using full CNN models with large training set sizes demonstrate a marked improvement in the AUC to 0.97, with a sensitivity and specificity of 91% and 92%, respectively [26]. More recently, ensemble models created using a variety of CNN architectures based on 990 plain abdominal radiographs showed an AUC of 0.96, corresponding to a sensitivity and specificity of 91 and 93%, respectively, in identifying small bowel obstruction [20]. Considering that abdominal radiographs are less sensitive than CT for the diagnosis of small bowel obstruction, similar studies using CT images are warranted. Post-validation, the development of such AI-driven systems could alert clinicians to the presence of critical clinical features warranting expedited clinical review and thereby improve patient outcomes.

### 3.2. Intussusception

In children aged 3 to 36 months old, intussusception remains the most common cause of intestinal obstruction [27]. While roughly 84% of patients experience alleviation of symptoms when diagnosed and treated with an air enema within 24 h of onset, delays in treatment can result in complications such as ischemia, necrosis, and perforation [27]. Traditionally, abdominal radiographs have markedly low sensitivity for detection of intussusception (<50%) and a poor rate of interobserver reliability [28]. In recent years, however, studies investigating risk stratification for intussusception in children have demonstrated the utility of abdominal X-rays as an initial diagnostic modality, with one study reporting sensitivity and specificity values of 0.77 and 0.79, respectively [29]. Unlike ultrasound (US), plain radiography is unaffected by operator skill and equipment variability and remains an inexpensive option for a first-line screening test [19]. As such, the implementation of AI algorithms in abdominal radiography may have a broad patient impact, and it shows promise as an initial point of entry.

The recent studies of AI applications in the detection of intussusception have focused on the implementation of deep learning algorithms for abdominal radiographs and have indicated that the technique may add value to the field. In one retrospective study of 681 pediatric patients, including 242 children diagnosed with intussusception, the authors used a You Only Look Once, Version 3 (YOLOv3) deep learning algorithm to validate automated detection [30]. The sensitivity of the algorithm was higher when compared with radiologist interpretation alone (0.76 vs. 0.46), while there were no significant differences in the specificity (0.96 vs. 0.92) [30]. More recent studies with larger sample sizes have demonstrated improved detection of intussusception with ranges between 0.91–0.94 and 0.85–0.91, respectively [19]. Other authors have described similar findings with AUC values of 0.95 and 0.97 and an accuracy of 0.93 and 0.95 [19]. Thus, as more data are gathered, hospitals may train these algorithms and institute them for routine use in emergency radiology.

### 3.3. Acute Appendicitis

Acute appendicitis is one of the most common causes of acute abdominal pain in the emergency department [31]. However, many patient-specific factors make detection of the appendix and diagnosis of appendicitis difficult, such as unusual appendix location, scanty intraabdominal fat, prominent cecal wall thickening, and abscess formation adjacent to the adnexa [31]. While both US and CT are important in the diagnosis of acute appendicitis, CT is considered the gold-standard diagnostic tool as it circumvents the issues of operator dependency, abundant bowel gas, and obesity that are prevalent in ultrasound techniques [31]. As for AI applications for the detection of acute appendicitis, the literature remains sparse, with only one study investigating the performance of the CNN-based diagnosis algorithm for abdominopelvic CT imaging [18]. In this retrospective multicenter study, the authors obtained a total of 667 image sets from 215 acute appendicitis patients and 452 controls for the algorithm training [18]. Following training, the CNN algorithm achieved a diagnostic accuracy of 91.5% for all image sets, with a reported sensitivity and specificity of 90.2% and 92.0%, respectively [18].

Although the diagnostic performance of the CNN algorithm was excellent, many false negatives were reported as the AI algorithm oftentimes misinterpreted early phase acute appendicitis, appendiceal perforation with abscess, and small mesenteric fat [18]. Some cases of false negatives were difficult to comprehend as trained humans never deemed these cases as normal [18]. Thus, the application of a CNN-based diagnosis algorithm in CT imaging may be useful in conjunction with a trained radiologist. Along with a thorough examination for false negatives, CNN-based acute appendicitis detection could potentially be implemented as a second opinion in order to improve diagnostic accuracy acutely.

More recently, a random forest-based predictive model of pediatric appendicitis was created and validated on a dataset obtained from 430 children and adolescents. The model used information extracted from patient history, clinical examination, laboratory parameters, and abdominal ultrasonography and reported areas under the precision-recall curve of 0.94, 0.92, and 0.70, respectively, for the diagnosis, management, and severity of appendicitis [21]. External validation using large sample sizes can increase the impact of such findings and help to identify and manage patients with potential appendicitis and its heterogeneous presentation in the pediatric population.

### 3.4. Colitis

Colitis is a chronic disease resulting from inflammation of the inner lining of the colon that can be caused by multiple etiologies, including ischemia, infection, neutropenia, and inflammatory bowel diseases (Crohn’s disease and ulcerative colitis) [32]. In the acute setting, patients present with diarrhea and abdominal pain, and CT is frequently utilized to evaluate patients for the presence of this disease [33]. Certain CT findings, such as wall thicknesses greater than 3 mm and the presence of an “accordion sign” (due to trapping of oral contrast between thickened haustral folds and mucosal ridges), are considered to be representative of colitis [33]. In addition, both of these radiographic findings can serve as important imaging markers for the AI-based detection of colitis. While older studies investigated the use of traditional AI methods, such as hand-crafted features (Gabor filters) and support vector machines to detect and classify colitis, these methods rely on expert knowledge and the segmentation of muscle, kidneys, and liver to reduce false-positive classification [33]. Other strategies such as high-capacity, region-based CNN have also demonstrated utility in colitis screening, with some studies reporting sensitivities and specificities as high as 94% and 95%, respectively [17,34]. These models have observed AUC values as high as 0.99 and are encouraging for potential clinical application [34].

In a multicenter diagnostic study involving five hospitals in China, deep learning models constructed from 49,154 colonoscopy images collected 1772 participants with inflammatory bowel disease (IBD) and normal controls; the identification accuracy obtained by the deep learning model was superior to that of experienced endoscopists per patient (deep model vs. trainee endoscopist, 99.1% vs. 78.0%) and per lesion (deep model vs. trainee endoscopist, 90.4% vs. 59.7%) [22]. While the difference between the two approaches was smaller when an experienced endoscopist was included, the deep learning still performed significantly (*p* < 0.001) better than its visual assessment-based counterpart.

## 4. Trauma

### 4.1. Hemoperitoneum

In the setting of trauma, point of care ultrasound (POCUS), particularly the Focused Assessment with Sonography for Trauma (FAST) examination, is the gold standard for rapid detection of hemoperitoneum [18]. Certain sonographic findings, such as free fluid in the right upper quadrant (RUQ), are the most important independent predictors of therapeutic laparotomy in trauma [15,35,36,37]. Positive free fluid findings on US imaging can also narrow the differential diagnosis and aid decision making for antibiotic administration, surgery, or transfer of care to tertiary referral hospitals [15,38,39,40].

As the demand for on-call imaging expands, the need for efficient and accurate imaging in POCUS has led to research developments in the feasibility of automated detection systems. In one retrospective pilot study by Gwin et al., the authors employed cross-sectional RUQ views from FAST examinations to investigate the feasibility of automating free fluid detection [15]. A traditional AI algorithm was developed with features related to geometric properties (i.e., linearity, curvilinearity, radius angle covariance, roundness, position, and area), grayscale color properties of shape (i.e., echogenicity, echo variability, medial/lateral neighborhood echogenicity, and medial/lateral neighborhood variability), edge sharpness, and pixilation [15]. The features were subsequently inputted into a support vector machine for the classification of hypoechoic regions of interest as ‘free fluid’ or ‘not free fluid’. This study reported a sensitivity and specificity of 100% and 90%, respectively, in detecting free fluid on FAST examination for trauma; these values are within range of those reported in studies evaluating the human interpretation of free fluid detection. The authors also concluded that AI applications may also allow for the expedited identification of abdominal free fluid in the acutely ill non-trauma patient [15]. Ultimately, these results warrant further investigation and applications in other disease states, as well as the expansion of the approach to all quadrants for true improvements in clinical utility. Furthermore, implementation of automated detection systems may help reduce unnecessary patient transfers to tertiary care centers and make for an ideal triage tool. Taken together, automated detection systems may be vital in reducing the burden of imaging interpretation volumes for the on-call radiologist.

More recently, a multiscale deep learning approach designed for the quantitative visualization of traumatic hemoperitoneum using CT images showed a significantly improved performance (accuracy of 84%, sensitivity of 82%, specificity of 93%, positive predictive value of 86%, and negative predictive value of 83%) for the prediction of a composite outcome of surgical or angiographic hemostatic intervention, massive transfusion, and mortality compared with that of the conventional volume estimation methods [23]. Similar studies using larger sample sizes, multicentric data, and the inclusion of negative controls can improve the impact of the findings and support the development of clinical aids to rapidly and objectively quantify hemoperitoneum.

### 4.2. Traumatic Pelvic Injuries

These automated detection techniques may also be valuable in the identification of traumatic pelvic injuries as rapid detection remains crucial to the timely delivery of life-saving interventions [41]. Recent studies indicate that approximately 22% of patients with pelvic injuries have concomitant abdominal trauma [42]. Of special significance is the fact that pelvic fractures are a marker of injury from major force and are associated with morbidity and mortality from bleeding and abdominal compartment syndrome, as well as intraabdominal abscesses [9,43,44].

Supervised learning has commonly been used to detect fractures in local regions and has demonstrated an accuracy comparable to that of physicians [45]. Deep learning studies report an accuracy upwards of 90% for detecting hip fractures in various settings [46,47]. Recently, Cheng et al. reported a scalable physician-level deep learning algorithm (PelviXNet) that detects universal trauma on pelvic radiographs. Using data from 5204 pelvic radiographs, PelviXNet yields an AUC of 0.97 (95% CI, 0.96–0.98) [24]. While the results are valuable, most of the conditions analyzed in this study are rarely missed by physicians, leading to a limitation in its impact [48]. Multicentric studies using large sample sizes, particularly those using data including more complex injuries which are visually difficult to discern, will be very impactful to the clinical community.

In severe pelvic fractures, injuries to the bladder are most common (15%) followed by the liver (10%) [9]. In milder pelvic fractures, the most commonly injured organ is the liver (6%) [9]. While contrast-enhanced CT is the gold-standard diagnostic test for pelvic trauma, the sensitivity of this technique in evaluating both mild and severe pelvic fractures is only 66% [49]. This can be attributed to a multitude of imaging complexities, including low resolution, noise, partial volume effects, and inhomogeneities, which are particularly relevant in identifying mild/small bone fractures [41]. These irregularities render image labeling difficult, often requiring multiple reads to confirm the existence and details of a fracture. Thus, computer-assisted support may have a potential niche in assisting emergency radiologists in making accurate diagnoses and assessing the severity of pelvic fractures with shorter turnaround times.

Unfortunately, the literature surrounding AI algorithms for the CT detection of pelvic fractures remains sparse. One retrospective study investigated the feasibility of automated fracture detection in 12 patients, including 8 patients presenting with mild and small fractures [41]. The authors developed a traditional AI algorithm involving pelvic bone segmentation through registered active shape models, adaptive window creations, 2D stationary wavelet transformations, masking, and boundary tracing [41]. The proposed model reduced the overall processing speed and achieved a 92% accuracy, 93% sensitivity, and 89% specificity in detecting pelvic bone fractures [41]. Furthermore, this model quantified certain fracture features, such as separation distance and angle, that are not visible to the human eye.

Computer-assisted decision support for CT can also be implemented in the automated segmentation and measurement of traumatic pelvic hematomas. While the volume of pelvic hematoma is the strongest independent predictor of arterial injury needing angioembolization in trauma patients with pelvic fractures, the measurement of pelvic hematoma volumes through current methods (e.g., semiautomated seeded region growing) are time-consuming [14]. In addition, the shape and location of pelvic hematomas are often variable and have poorly defined margins, further muddling detection. Thus, hospitals may benefit from more efficient automated approaches. In a retrospective study of 253 trauma patients, Dreizin et al. assessed the performance of a deep learning algorithm for the automated segmentation and measurement of pelvic hematoma volume [14]. Not only did this algorithm contain a recurrent saliency transformation network, but it also made objective volumetric hematoma measurements for the prediction of arterial injury requiring angioembolization. Ultimately, these authors reported that the aggregate measure of performance for the model achieved an area-under-curve (AUC) of 0.81, which is comparable to manual measurements of pelvic hematoma volume (AUC of 0.80) [14]. Other studies reported similar findings and noted that the use of deep learning algorithms for hematoma measurements demonstrated an improved prediction of the need for pelvic packing, massive transfusion, and in-hospital mortality when compared to subjective hematoma measurements [50]. Thus, the optimization of hematoma measurement through AI could augment outcome prediction for trauma patients and may guide treatment planning for emergency radiologists.

## 5. Abdominal Aortic Aneurysms

Abdominal aortic aneurysms (AAAs) are a life-threatening disease characterized by segmental weakening and ballooning of the aorta [51]. While the only curative treatment of AAA is open or endovascular repair, the decision to proceed with surgical repair requires careful consideration of the surgical risks and the risk of aneurysm rupture. Thus, CT imaging is often utilized for operative planning as it allows visualization of the aorta, access vessels, aneurysm morphology, and coexistent occlusive disease [51]. In recent years, AI methods have been proposed to improve the efficiency of image segmentation, the detection of AAA, and the characterization of AAA geometry and fluid dynamics.

A recent systematic review described 15 studies of AI methods on the segmentation of the abdominal aorta [51]. Manual segmentation is time-consuming, often requiring 30 min and is operator-dependent [51]. To reduce segmentation time and the reliability of segmentation, one approach utilized an active shape model (ASM) segmentation scheme for CT angiography (CTA) images [52]. This technique refers to the development of a statistical shape model derived from labeled landmark points and iteratively fitted to an image. Following manual segmentation of the first slice, a shape model of the contours in adjacent image slices is iteratively fitted over the entire volume of the AAA. This in turn reduces the time required for expert segmentation by a factor of six. Other potential techniques include semiautomatic approaches for segmentation with the use of a 3D deformable model and level-set algorithms [53].

AI methods have also been proposed to quantify the morphologic aspects of AAA. In a study by Zhuge et al., predefined features of intensity, volume, and aorta shape from 20 CTA scans of AAA patients were utilized to train a support vector machine classifier [54]. Following preprocessing, global region analysis, surface initialization, local feature analysis, and level set segmentation, the authors observed the mean and worst-case values of the volume overlap at 95% and 93% [54]. The mean segmentation time was also reduced from 30 min to 7.4 min. Other studies have employed finite-element, analysis-based approaches to automate the analysis of CT and magnetic resonance imaging (MRI) images [55]. These applications have been extended to multimodal imaging using neural network fusion models [56]. In this setting, AI models allow a shared representation of the aorta in both the CT and the MRI images. In addition to aneurysm shape, both intraluminal thrombus and calcifications contribute to the development of AAA and the risk of rupture [57]. Recent studies have employed fully automated pipelines to detect the aortic lumen and characterize the intraluminal thrombus and calcifications with computational times of <1 min [58].

As the precise characterization of AAA geometry and arterial wall thickness is vital for the assessment of the rupture risk, several studies have investigated the development of neural network algorithms for accurate measurement [16,59]. One study reported an association between AI performance and the manual assessment performed by vascular surgeons, with coefficients of variation of 11% for ruptured AAA and 13% for non-ruptured AAA [59]. In another study, the authors developed a decision tree algorithm from 76 contrast-enhanced CT scans to characterize AAA geometry into 25 sizes and shapes [16]. Ultimately, this model yielded a prediction accuracy of 87% [16].

AI techniques have also been utilized in the characterization of AAA fluid dynamics as wall shear stress also accounts for AAA rupture risk [60]. While some studies have measured computational fluid dynamics and estimated wall shear stress from geometric parameters, other authors have utilized machine learning to calculate wall shear stress and predict wall shear stress distribution in carotid bifurcation models [60,61,62]. These studies demonstrate the potential clinical utility of AI in distinguishing AAA morphology and may be effective in reducing the costs associated with image analysis.

Using multicenter, multi-scanner, multiphase CT data, the 3D ResNet model demonstrated a high performance (AUC of 0.95) for fully automated abdominal aneurysm detection in an abdominal CT scan [25]. While promising, the study was conducted on a small patient cohort of 187 CT scans as a training dataset, potentially limiting the dataset variability and, thus, the generalizability. The validation of similar approaches in a larger cohort can increase the robustness of the findings and ultimately aid the transition of such AI-driven workflows into clinical practice.

## 6. Practical Applications

In recent years, a number of promising AI algorithms have been developed for use in the emergency setting. In a study by Kim et al., the authors tested the accuracy of artificial intelligence and deep learning-based algorithms in the context of diagnosing ileocolic intussusception on abdominal radiographs in the pediatric population [63]. Ultimately, the deep learning-based algorithms provided higher sensitivity in diagnosing intussusception in children under five years old when compared to clinical radiologists (0.76 vs. 0.46, *p* =  0.013), but demonstrated no statistical difference in specificity (0.96 vs. 0.92, *p*  =  0.32) [30]. Pang et al. also utilized the Yolov3-arch neural network in the clinical setting by identifying cholelithiasis and classifying gallstones on CT images [63]. This algorithm was applied to a medical image dataset comprising 223,846 CT images, with gallstones present in 1369 patients. The diagnostic accuracy of this algorithm was ultimately reported to be 86.5%, thus indicating the practical use of AI in assisting radiologists in gallstone detection [63].

## 7. Discussion

Despite these strides, several barriers remain that prevent the clinical translation of AI techniques into daily workflow [5,64,65]. These include both ethical and medicolegal challenges, such as standardization difficulties across multiple centers, potential disagreement between radiologists and AI, and gaining trust in the black-box deep learning approach [12]. Some of the implementation-related challenges include incorporating AI within PACS and EMR systems, determining the level of AI–human interaction, and packaging these algorithms into a widely acceptable product [66].

Beyond the implementation barriers, there are deep-rooted issues with artificial intelligence at its core. First, most machine learning technologies have high sensitivity but low to moderate specificity [2]. Thus, AI can be highly beneficial as a screening tool, but oftentimes falls short when ruling on a diagnosis, particularly when dealing with overlapping structures. For instance, an algorithm may identify a micro-calcification smaller than the human eye as nephrolithiasis, which could actually be an early atherosclerotic plaque in the vessel running posterior to the organ of interest. These issues become further compounded by the fact that this novel technology does not consider the full clinical picture when making a diagnosis. As many gastrointestinal pathologies can present similarly on imaging, it is imperative to consider patient demographics and history. For example, while AI-based imaging may correctly identify an adrenal nodule, the clinical context of episodic hypertension and tachycardia would favor a diagnosis of pheochromocytoma, whereas a patient with new-onset truncal obesity, insulin resistance, and hirsutism most likely has an adrenocortical adenoma [67]. In addition, a renal hyperdensity can be interpreted as active extravasation in the context of trauma or nephrolithiasis in the context of unilateral flank pain and colicky pain radiating to the groin. Thus, machine learning techniques should not be utilized as a stand-alone technology, but instead applied under the supervision of a trained radiologist.

For successful implementation of deep learning systems in radiology, large well-annotated datasets of medical images are needed to detect subtle differences in disease states [4]. Yet, a scarcity of this large-scale data currently exists [68]. For medical image datasets that are too small to generate vast networks, pre-trained deep learning networks obtained from large-scale natural images may be repurposed and transferred over, in a process known as transfer learning. However, these techniques are fraught with limitations. Of the datasets that have been generated, many inaccuracies have been identified, particularly in patients who had undergone long periods of hospitalization. In a study by Behzadi-Khormouji et al. that conducted a quality control of various AI datasets, the authors noted that certain CXRs labeled as “no interval change”, were incorrectly coded as “no finding” within the dataset, and thus, were being utilized as a standardized normal [2]. Consequently, the “high accuracy rate” associated with AI models may actually be due to inaccurate training/coding, yielding unforeseen errors [2]. Thus, machine learning technologies require constant retraining and evaluation to ensure their accuracy and precision and that they stay current with the constant learning curve present in medicine.

Additional challenges associated with this technology include the poor generalizability of models trained on one dataset (single-institution dataset) to other data [69,70]. Due to the high-risk nature of translating AI technologies developed from single institutions to widespread clinical practice, governing bodies such as the US Food and Drug Administration (FDA) have attempted to adopt specific regulatory frameworks to ensure effective safeguards. To date, these frameworks have cleared medical devices utilizing “locked” algorithms, i.e., those that provide reproducible results with the same inputs. Changes beyond the original market authorization for these algorithms would require FDA premarket review [71]. However, artificial intelligence/machine learning (AI/ML)-based medical devices increasingly utilize deep learning networks that adapt over time, where the adaptation or change is only recognized after distribution. Current regulatory frameworks have not been designed for medical devices using these adaptive algorithms.

Distributional shift can also greatly impact AI technology and lead to erroneous predictions [72]. For example, models can appear to perform with high accuracy but may fail if the dataset suddenly shifts. As disease patterns are constantly changing, a mismatch can occur between the training and the operational data [72]. In order to combat this, the FDA proposed a potential solution to this problem where manufacturers can submit periodic updates and real-world performance monitoring to the FDA as part of an algorithm-change protocol [71]. This method falls under the framework of a total product lifecycle regulatory approach, allowing the integration of pre-market and post-market surveillance data for medical devices using AI/ML-based technologies. Within the field of radiology, 21 AI/ML-based algorithms are FDA-approved as medical devices, 3 of which are used for CT-based lesion detection (Arterys Oncology DL, Arterys MICA, and QuantX), 2 of which are used for stroke and hemorrhage detection (ContaCT, Accipiolx, and Icobrain), 6 of which are deep learning algorithms used to improve image processing (SubtlePET, Deep Learning Image Reconstruction, Advanced Intelligent Clear-IQ Engine, SubtleMR, and AI-Rad Companion), and 4 of which are focused on acute care for pneumothorax, wrist fracture diagnosis, and triage of head, spine, and chest injuries (Health PNX, Critical Care Suite, OsteoDetect, and Aidoc Medical BriefCase System) [72]. Deep learning algorithms developed from single institutions will require approval under these regulatory pathways for widespread clinical application. Ultimately, this approach can help the FDA embrace the iterative improvement power of AI/ML-based technologies as medical devices, while simultaneously ensuring patient safety.

## 8. Conclusions

The field of emergency radiology can greatly benefit from AI applications in image segmentation, automated detection, and outcome prediction for a variety of abdominopelvic pathologies. Not only can AI algorithms automatically identify subtle disease states and provide quantitative characterization of disease severity, but they also have the potential to improve workflow efficiency and reduce overall workloads. In addition, AI can help augment human decision making and serve as a second opinion in complicated cases. As most AI methods are trained in one specific task, it remains to be seen whether AI will be broadly implemented in the detection of multiple abdominopelvic pathologies, as outlined here. While the field of AI in emergency radiology is expanding exponentially, many challenges exist that hinder the clinical translation of these technologies.

## Figures and Tables

**Table 1 diagnostics-12-01351-t001:** Overview of Deep Learning Algorithms Developed For Use in the Emergency and Clinical Setting.

Title/Author	Journal/Year/Type	Data	Data Processing	Application	Model	Performance	Reference
Pelvic Fractures: Epidemiology and Predictors of Associated Abdominal Injuries and Outcomes Demetriades et al. [9]	*J. Am. Coll. Surg.* 2002 *Original*	No DL					Demetriades D, et al. Pelvic fractures: epidemiology and predictors of associated abdominal injuries and outcomes. J Am Coll Surg. 2002 Jul;195(1):1–10. doi:10.1016/s1072-7515(02)01197-3. PMID: 12113532.
Detecting pelvic fracture on 3D-CT using deep convolutional neural networks with multi-orientated slab images Ukai et al. [10]	*Scientific Reports* 2021 *Original*	Multisource CT images acquired from 93 subjects who had one or more pelvic fractures. Multisource CT images acquired from 112 subjects identified by orthopedic surgeons as not having any fractures.	Voxel size and Intensity range harmonization	Automatically detect pelvic fractures from pelvic CT images of an evaluating subject.	DCNN: YOLOv3	Area under the curve (AUC) was 0.824, with 0.805 recall and 0.907 precision.	Ukai K, et al. Detecting pelvic fracture on 3D-CT using deep convolutional neural networks with multi-orientated slab images. Sci Rep 11, 11716 (2021). https://doi.org/10.1038/s41598-021-91144-z.
Accuracy of Abdominal Radiography in Acute Small-Bowel Obstruction: Does Reviewer Experience Matter? Thompson et al. [11]	*Abdominal Imaging* 2007 *Original*	No DL					Thompson WM, et al. Accuracy of abdominal radiography in acute small-bowel obstruction: does reviewer experience matter? AJR Am J Roentgenol. 2007 Mar;188(3):W233-8. doi:10.2214/AJR.06.0817. PMID: 17312028.
Abdominal Radiography Findings in Small-Bowel Obstruction: Relevance to Triage for Additional Diagnostic Imaging Lappas et al. [12]	*AJR* 2001 *Original*	No DL					Lappas JC, et al. Abdominal radiography findings in small bowel obstruction: relevance to triage for additional diagnostic imaging. AJR 2001; 176:167–174.
Detection of high-grade small bowel obstruction on conventional radiography with convolutional neural networks Cheng et al. [13]	*Ab. Radiol.* 2018 *Original*	3663 supine abdominal radiographs	Pixel size and Intensity range harmonization	Determine whether a deep CNN can be trained with limited image data to detect high-grade small bowel obstruction patterns on supine abdominal radiographs.	Inception v3 CNN	The neural network achieved an AUCof 0.84 on the test set (95% CI 0.78–0.89). At the maximum Youden index (sensitivity + specificity-1), the sensitivity of the system for small bowel obstruction was 83.8%, with a specificity of 68.1%.	Cheng PM, et al. Detection of high-grade small bowel obstruction on conventional radiography with convolutional neural networks. Abdom Radiol (NY) 2018;43(5):1120–1127.
Performance of a Deep Learning Algorithm for Automated Segmentation and Quantification of Traumatic Pelvic Hematomas on CT Dreizin et al. [14]	*Journal of Digital Imaging* 2021 *Original*	253 C/A/P admission trauma CT	Pixel size and Intensity range harmonization	Determine if RSTN would result in sufficiently high Dice similarity coefficients to facilitate accurate and objective volumetric measurements for outcome prediction (arterial injury requiringangioembolization).	Recurrent Saliency Transformation Network vs. 3D U-Net	Dice scores in the test set were 0.71 (SD ± 0.10) using RSTN, compared to 0.49 (SD ± 0.16) using a baseline Deep Learning Tool Kit (DLTK) reference 3D U-Net architecture.	Dreizin D, et al. “A Multiscale Deep Learning Method for Quantitative Visualization of Traumatic Hemoperitoneum at CT: Assessment of Feasibility and Comparison with Subjective Categorical Estimation.” Radiology. Artificial intelligence vol. 2,6 e190220. 11 Nov. 2020, doi:10.1148/ryai.2020190220.
Image Segmentation and Machine Learning for Detection of Abdominal Free Fluid in Focused Assessment With Sonography for Trauma Examinations A Pilot Study Sjogren et al. [15]	*J. Ultrasound Med.* 2016 *Original*	20 cross-sectional abdominal US videos (FAST)	None	Test the feasibility of automating the detection of abdominal free fluid in focused assessment with sonography for trauma (FAST) examinations.	ML: SVM	The sensitivity and specificity (95% confidence interval) were 100% (69.2–100%) and 90.0% (55.5–99.8%), respectively.	Sjogren AR, et al. “Image Segmentation and Machine Learning for Detection of Abdominal Free Fluid in Focused Assessment With Sonography for Trauma Examinations: A Pilot Study.” Journal of ultrasound in medicine: official journal of the American Institute of Ultrasound in Medicine vol. 35,11 (2016): 2501–2509. doi:10.7863/ultra.15.11017.
Quantitative Assessment of Abdominal Aortic Aneurysm Geometry Shum et al. [16]	*Ann. Biomed. Eng.* 2011 *Original*	76 CTs of patients with aneurysms	None	Test the feasibility that aneurysm morphology and wall thickness are more predictive of rupture risk and can be the deciding factors in the clinical management.	ML: Decision Tree	The model correctly classified 65 datasets and had an average prediction accuracy of 86.6% (κ = 0.37).	Shum J, et al. “Quantitative assessment of abdominal aortic aneurysm geometry.” Annals of biomedical engineering vol. 39,1 (2011): 277–286. doi:10.1007/s10439-010-0175-3.
Detection and Diagnosis of Colitis on Computed Tomography Using Deep Convolutional Neural Networks Liu et al. [17]	*Med Phys.* 2017 *Original*	CT images of 80 patients with colitis	None	Develop deep convolutional neural networks methods for lesion-level colitis detection and a support vector machine (SVM) classifier for patient-level colitis diagnosis on routine abdominal CT scans.	Faster Region-based Convolutional Neural Network (Faster RCNN) with ZF net	For patient-level colitis diagnosis, with ZF net, the average areas under the ROC curve (AUC) were 0.978 ± 0.009 and 0.984 ± 0.008 for RCNN and Faster RCNN method, respectively.	Liu J, et al. Detection and diagnosis of colitis on computed tomography using deep convolutional neural networks. Med Phys 2017;44(9):4630–4642.
Convolutional-neural-network-based diagnosis of appendicitis via CT scans in patients with acute abdominal pain presenting in the emergency department Park et al. [18]	*Scientific Reports* 2020 *Original*	667 CT image sets from 215 patients with acute appendicitis and 452 patients with a normal appendix	Data augmentation to prevent over-fitting	Test feasibility of a neural network-based diagnosis algorithm of appendicitis by using CT for patients with acute abdominal pain visiting the emergency room (ER).	Deep CNN	Diagnostic performance was excellent inboth the internal and external validation with an accuracy larger than 90%.	Park JJ, et al. Convolutional-neural-network-based diagnosis of appendicitis via CT scans in patients with acute abdominal pain presenting in the emergency department. Sci Rep. 2020 Jun 12;10(1):9556. doi:10.1038/s41598-020-66674-7. PMID: 32533053; PMCID: PMC7293232.
Deep learning algorithms for detecting and visualizing intussusception on plain abdominal radiography in children: a retrospective multicenter study Kwon et al. [19]	*Scientific Reports* 2021 *Original*	9935 X-rays	None	Verify a deep CNN algorithm to detect intussusception in children using a human-annotated dataset of plain abdominal X-rays.	Single Shot MultiBox Detector and ResNet	The internal test values after training with two hospital datasets were 0.946 to 0.971 for the area under the receiver operating characteristic curve (AUC), 0.927 to 0.952 for the highest accuracy, and 0.764 to 0.848 for the highest Youden index.	Kwon G, et al. Deep learning algorithms for detecting and visualising intussusception on plain abdominal radiography in children: a retrospective multicenter study. Sci Rep 10, 17582 (2020). https://doi.org/10.1038/s41598-020-74653-1.
An artificial intelligence deep learning model for identification of small bowel obstruction on plain abdominal radiographs Kim et al. [20]	*British Journal of Radiology* 2021 *Original*	990 plain abdominal radiographs	None	Detect small bowel obstructions of plain abdominal X-rays.	VGG16, Densenet121, NasNetLarge, InceptionV3, and Xception	The model showed an AUC of 0.961, corresponding to sensitivity and specificity of 91 and 93%, respectively.	Kim DH, et al. “An artificial intelligence deep learning model for identification of small bowel obstruction on plain abdominal radiographs.” The British journal of radiology vol. 94,1122 (2021): 20201407. doi:10.1259/bjr.20201407.
Performance of deep learning-based algorithm for detection of ileocolic intussusception on abdominal radiographs of young children Kim et al. [19]	*Scientific Reports* 2019 *Original*	Abdominal radiographs of 681 children	Intensity normalization using z-score	Detect ileocolic intussusception on abdominal radiographs of young children.	YOLO v3	The sensitivity of the algorithm was higher compared with that of the radiologists (0.76 vs. 0.46, *p* = 0.013), while specificity was not different between the algorithm and the radiologists (0.96 vs. 0.92, *p* = 0.32).	Kim S, et al. Performance of deep learning-based algo-rithm for detection of ileocolic intussusception on abdominal radiographs of young children. Sci Rep. 2019 Dec 19;9(1):19420. doi:10.1038/s41598-019-55536-6. PMID: 31857641; PMCID: PMC6923478.
Using Machine Learning to Predict the Diagnosis, Management and Severity of Pediatric Appendicitis Marcinkevics et al. [21]	*Frontiers in Pediatrics* 2021 *Original*	430 children and adolescents	None	Detect pediatric appendicitis.	Logistic regression, random forests, and gradient boosting machines	A random forest classifier achieved areas under the precision-recall curve of 0.94, 0.92, and 0.70, respectively, for the diagnosis, management, and severity of appendicitis.	Marcinkevics R, et al. (2021). Using Machine Learning to Predict the Diagnosis, Management and Severity of Pediatric Appendicitis [Original Research]. Frontiers in Pediatrics, 9. https://doi.org/10.3389/fped.2021.662183.
Development and Validation of a Deep Neural Network for Accurate Identification of Endoscopic Images From Patients With Ulcerative Colitis and Crohn’s Disease Ruan et al. [22]	*Frontiers in Medicine* 2022 *Original*	49,154 colonoscopy images from 1772 patients	Data augmentation using operations such as horizontal flipping, vertical flipping, random cropping, random rotation, brightness adjustment, contrast adjustment, and saturation adjustment, CutMix algorithm	Detect ulcerative colitis and Crohn’s disease using endoscopic images.	ResNet50	The identification accuracy achieved by the deep learning model was superior to that of experienced endoscopists per patient (deep model vs. trainee endoscopist, 99.1% vs. 78.0% and per lesion (deep model vs. trainee endoscopist, 90.4% vs. 59.7%. While the difference between the two was lower when an experienced endoscopist was included, the deep learning still performed significantly (*p* < 0.001) better.	Ruan G, et al. (2022). Development and Validation of a Deep Neural Network for Accurate Identification of Endoscopic Images From Patients With Ulcerative Colitis and Crohn’s Disease [Original Research]. Frontiers in Medicine, 9. https://doi.org/10.3389/fmed.2022.854677.
A Multiscale Deep Learning Method for Quantitative Visualization of Traumatic Hemoperitoneum at CT: Assessment of Feasibility and Comparison with Subjective Categorical Estimation Dreizin et al. [23]	*Radiology AI* 2020 *Original*	CT images of 130 patients	Pixel size and Intensity range harmonization	Evaluate the feasibility of a multiscale deep learning algorithm for quantitative visualization and measurement of traumatic hemoperitoneum and compare diagnostic performance for relevant outcomes with categorical estimation.	MSAN TensorFlow	AUCs for automated volume measurement and categorical estimation were 0.86 and 0.77, respectively (*p* = 0.004). An optimal cutoff of 278.9 mL yielded accuracy of 84%, sensitivity of 82%, specificity of 93%, positive predictive value of 86%, and negative predictive value of 83%.	Dreizin D, et al. “A Multiscale Deep Learning Method for Quantitative Visualization of Traumatic Hemoperitoneum at CT: Assessment of Feasibility and Comparison with Subjective Categorical Estimation.” Radiology. Artificial intelligence vol. 2,6 e190220. 11 Nov. 2020, doi:10.1148/ryai.2020190220.
A scalable physician-level deep learning algorithm detects universal trauma on pelvic radiographs Cheng et al. [24]	*Nat. Comm.* 2021 *Original*	5204 pelvic radiographs	Zero-padding and resizing, Data augmentation such as translation, flipping, scaling, rotation, brightness and contrast	Detect most types of trauma-related radiographic findings on pelvic radiographs.	PelviXNet	PelviXNet yielded an area under the receiver operating characteristic curve (AUROC) of 0.973 (95% CI, 0.960–0.983) and an area under the precision-recall curve (AUPRC) of 0.963 (95% CI, 0.948–0.974) in the clinical population test set of 1888 PXRs. The accuracy, sensitivity, and specificity at the cutoff value were 0.924 (95% CI, 0.912–0.936), 0.908 (95% CI, 0.885–0.908), and 0.932 (95% CI, 0.919–0.946), respectively.	Cheng CT, et al. A scalable physician-level deep learning algorithm detects universal trauma on pelvic radiographs. Nat Commun 12, 1066 (2021). https://doi.org/10.1038/s41467-021-21311-3.
Automated Screening for Abdominal Aortic Aneurysm in CT Scans under Clinical Conditions Using Deep Learning Golla et al. [25]	*Diagnostics* (Basel) 2021 *Original*	187 heterogenous CT scans.	Pixel size and Intensity range harmonization, Data augmentation	Develop and validate an easily trainable and fully automated deep learning 3D AAA screening algorithm, which can run as a background process in the clinic workflow.	ResNet, VGG-16 and AlexNet	The 3D ResNet outperformed both other networks and achieved an accuracy of 0.953 and an AUC of 0.971 on the validation dataset.	Golla AK, et al. “Automated Screening for Abdominal Aortic Aneurysm in CT Scans under Clinical Conditions Using Deep Learning.” Diagnostics (Basel, Switzerland) vol. 11,11 2131. 17 Nov. 2021, doi:10.3390/diagnostics11112131.

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
