# Peer review of "An Extra Set of Intelligent Eyes: Application of Artificial Intelligence in Imaging of Abdominopelvic Pathologies in Emergency Radiology"

_diagnostics, 2022, doi:10.3390/diagnostics12061351_

Round 1

Reviewer 1 Report

The authors propose a review about the application of Artificial Intelligence in imaging of abdominopelvic pathologies in emergency radiology.

The search strategy for the reviewed works is unclear. The authors should consider turning the review into a systematic review following the PRISMA guidelines. There is for example a review about AI-based segmentation of the whole aortic vessel tree (also covering AAAs) that should be mentioned:

https://arxiv.org/abs/2108.02998

Furthermore, you should add figures and tables summarizing the pros/cons, metrics, used datasets, etc. for the reviewed works (please see the aortic vessel tree survey above).

Author Response

Thank you so much for your useful comments and suggestions. We went through and generated a table that summarized our metrics, search methods, and used datasets based on your suggestion. In addition, we worked on the organization of the paper and added more literature so that it was more comprehensive. Thank you for linking the AI-based segmentation of the whole aortic vessel tree paper. We made sure that our table modeled the tables present in the paper and also cited this article within our paper.

Reviewer 2 Report

This is a review article regarding application of artificial intelligence (AI) in abdominopelvic imaging in emergency radiology. The title is interesting but the content of this review is less appealing. The authors have subcategorized the application in three aspects namely the GI tract, trauma and abdominal aortic aneurysm. As stated by the authors, there are many limitations to the general applications. Not much new information regarding how AI can work in Emergency Radiology can be retrieved from this manuscript.

Author Response

Thank you so much for your comments and helpful suggestions. We went through and added multiple paragraphs of data from 2021/2022 to ensure that we are providing new information. In addition, we further expanded on our limitations section to try and make it more appealing to the reader.

Reviewer 3 Report

Thanks for giving me an opportunity to review this wonderful manuscript. The authors have described the topic nicely. However, I have several comments that need to be addressed before considering it for publication.

  1. Authors should describe artificial intelligence methods precisely in the overview selection.
  2.  Authors should provide information on which methods were used in the digestive tract, trauma, and abdominal aortic aneurysms classification. Which types of image modalities were used?.
  3.  Authors should write one paragraph regarding the limitations of existing AI methods in the discussion part.
  4. Please give future prospective on how we can use these models and results in the real-world clinical setting.

Author Response

Thank you so much for your valuable insight and suggestions.

  1. and 2: We created a table describing the AI methods utilized and put it in the overview section. In addition, we added information regarding classification of methodology for the primary papers utilized.

3. Multiple paragraphs on the limitations of existing AI methods have been added to the discussion.

4. We included multiple paragraphs on the most recent data regarding AI models and also included future prospective within the discussion section.

Reviewer 4 Report

Jeffrey Liu and coworkers reported a review article “An Extra Set of Intelligent Eyes: Application of Artificial Intelligence in Imaging of Abdominopelvic Pathologies in Emergency Radiology”. Here in this article, the authors described the novel artificial intelligence (AI) algorithms used in emergent image-based tasks, and discussed the challenges associated with the implementation of AI into clinical workflow. Overall, the review was well organized and discussed in a good manner. Therefore, I recommend the editor accept the manuscript with minor revision.

  • I suggest adding figures or tables for the cited literature. See the reported review articles “Diagnostics 2022, 12, 1006”.
  • The reference citation was mentioned in both roman letters and numbers format. It is confusing, I recommend modifying it according to the “Diagnostics” reference style.
  • Table 1 is missing in the MS.

Author Response

Thank you so much for useful suggestions and comments. We have created a large table to add to the manuscript per your suggestion. I have attached it to this reply for your review. In addition, the numbering of the references (numerical and roman numbers) were completed by the MDPI editor and not us. I believe it may be just a clerical error on their end. 

Round 2

Reviewer 1 Report

I thank the authors for addressing my comments and endorse the paper for publication in the current form.

Author Response

Thank you so much for your support. 

Reviewer 2 Report

This is a review article regarding application of artificial intelligence (AI) in abdominopelvic imaging in emergency radiology. The title is interesting but the content of this review is less appealing. The authors have subcategorized the application in three aspects namely the GI tract, trauma and abdominal aortic aneurysm. I appreciate very much the effort the authors have put in for the corrections and addition of multiple new paragraphs. Not much practical information regarding how AI can be applied in Emergency Radiology has been discussed in this manuscript.

Author Response

Thank you so much for your useful suggestions and comments. We added a section on Practical Applications of AI in the emergency setting based on your feedback. 

Reviewer 3 Report

Thank you for your revised version. The authors have improved a lot. It can be considered for publication now.

Author Response

Thank you so much for your support.